# spmodel: Spatial statistical modeling and prediction in R

**Michael Dumelle**[1]*, **Matt Higham**[2], **Jay M. Ver Hoef**[3]

**1** United States Environmental Protection Agency, Corvallis, Oregon, United States of America, **2** Department of Math, Computer Science, and Statistics, St. Lawrence University, Canton, New York, United States of America, **3** Marine Mammal Laboratory, National Oceanic and Atmospheric Administration Alaska Fisheries Science Center, Seattle, Washington, United States of America

* Dumelle.Michael@epa.gov

## Abstract

`spmodel` is an **R** package used to fit, summarize, and predict for a variety spatial statistical models applied to point-referenced or areal (lattice) data. Parameters are estimated using various methods, including likelihood-based optimization and weighted least squares based on variograms. Additional modeling features include anisotropy, non-spatial random effects, partition factors, big data approaches, and more. Model-fit statistics are used to summarize, visualize, and compare models. Predictions at unobserved locations are readily obtainable.

## Introduction

Spatial data are ubiquitous in everyday life and the scientific literature. As such, it is becoming increasingly important to properly analyze spatial data. Spatial data can be analyzed using a statistical model that explicitly incorporates the spatial dependence among nearby observations. Incorporating this spatial dependence can be challenging, but ignoring it often yields poor statistical models that incorrectly quantify uncertainty, impacting the validity of hypothesis tests, confidence intervals, and predictions intervals. `spmodel` provides tools to easily incorporate spatial dependence into statistical models, building upon commonly used **R** functions like `lm()`.

`spmodel` implements model-based inference, which relies on fitting a statistical model. Model-based inference is different than design-based inference, which relies on random sampling and estimators that incorporate the properties of the random sample [1]. [2] defines two types of spatial data that can be analyzed using model-based inference: point-referenced data and areal data (areal data are sometimes called lattice data). Spatial data are point-referenced when they are observed at point-locations indexed by x-coordinates and y-coordinates on a spatially continuous surface with an infinite number of locations. Spatial models for point-referenced data are sometimes called geostatistical models. Spatial data are areal when they are observed as part of a finite network of polygons whose connections are indexed by a neighborhood structure. For example, the polygons may represent counties in a state who are neighbors if they share at least one boundary. Spatial models for areal data are sometimes called spatial autoregressive models. For thorough overviews of model-based inference in a spatial context, see [2–4].

**Data Availability Statement:** The data used in this manuscript are available upon download of the spmodel R package from CRAN. To learn more, visit https://CRAN.R-project.org/package=spmodel. The manuscript also has a supplementary R package that contains all of the text, figures, and

code used in the manuscript's creation. To learn more, visit https://github.com/USEPA/spmodel. manuscript.

**Funding:** The author(s) received no specific funding for this work.

**Competing interests:** The authors have declared that no competing interests exist.

Several **R** packages exist on CRAN that analyze either point-referenced or areal spatial data. For point-referenced data, they include `fields` [5], `FRK` [6], `geoR` [7], `GpGp` [8], `gstat` [9], `LatticeKrig` [10], `R-INLA` [11], `rstan` [12], `spatial` [13], `spBayes` [14], and `spNNGP` [15]. For areal data, they include `brms` [16], `CARBayes` [17], `bigDM` [18], and `hglm` [19]. Unlike these aforementioned packages, `spmodel` is designed to analyze both point-referenced and areal data using a common framework and syntax structure. `spmodel` also offers many features missing from the aforementioned **R** packages—together in one **R** package, `spmodel` offers detailed model summaries, extensive model diagnostics, non-spatial random effects, anisotropy, big data methods, prediction, the option to fix spatial covariance parameters at known values, and more.

The rest of this article is organized as follows. We first give a brief theoretical introduction to spatial linear models. We then outline the variety of methods used to estimate the parameters of spatial linear models. Next we explain how to obtain predictions at unobserved locations. Following that, we detail some advanced modeling features, including random effects, partition factors, anisotropy, and big data approaches. Finally we end with a short discussion.

Before proceeding, we install `spmodel` from CRAN and load it by running

```
R> install.packages("spmodel")
R> library(spmodel)
```

We create visualizations using ggplot2 [20], which we install from CRAN and load by running

```
R> install.packages("ggplot2")
R> library(ggplot2)
```

We also show code that can be used to create interactive visualizations of spatial data with `mapview` [21]. `mapview` has many backgrounds available that contextualize spatial data with topographical information. Before running the `mapview` code provided interactively, make sure that `mapview` is installed and loaded.

`spmodel` contains various methods for generic functions defined outside of `spmodel`. To find relevant documentation for these methods, run `help("generic.spmodel", "spmodel")` (e.g., `help("fitted.spmodel", "spmodel")`, `help("summary. spmodel", "spmodel")`, `help("plot.spmodel", "spmodel")`, `help("predict.spmodel", "spmodel")`, `help("tidy.spmodel", "spmodel")`, etc.). We provide more details and examples regarding these methods and generics throughout this vignette. For a full list of `spmodel` functions available, see `spmodel`'s documentation manual.

## The spatial linear model

Statistical linear models are often parameterized as

$$\mathbf{y} = \mathbf{X}\boldsymbol{\beta} + \epsilon, \tag{1}$$

where for a sample size $n$, $\mathbf{y}$ is an $n \times 1$ column vector of response variables, $\mathbf{X}$ is an $n \times p$ design (model) matrix of explanatory variables, $\boldsymbol{\beta}$ is an $p \times 1$ column vector of fixed effects controlling the impact of $\mathbf{X}$ on $\mathbf{y}$, and $\epsilon$ is an $n \times 1$ column vector of random errors. We typically assume that $\mathrm{E}(\epsilon) = \mathbf{0}$ and $\mathrm{Cov}(\epsilon) = \sigma_\epsilon^2\mathbf{I}$, where $\mathrm{E}(\cdot)$ denotes expectation, $\mathrm{Cov}(\cdot)$ denotes covariance, $\sigma_\epsilon^2$ denotes a variance parameter, and $\mathbf{I}$ denotes the identity matrix.

The model in Eq 1 assumes the elements of $\mathbf{y}$ are uncorrelated. Typically for spatial data, elements of $\mathbf{y}$ are correlated, as observations close together in space tend to be more similar

than observations far apart [22]. Failing to properly accommodate the spatial dependence in **y** can lead researchers to incorrect conclusions about their data. To accommodate spatial dependence in **y**, an $n \times 1$ spatial random effect, $\boldsymbol{\tau}$, is added to Eq 1, yielding the model

$$\mathbf{y} = \mathbf{X}\boldsymbol{\beta} + \boldsymbol{\tau} + \epsilon, \tag{2}$$

where $\boldsymbol{\tau}$ is independent of $\epsilon$, $\mathrm{E}(\boldsymbol{\tau}) = \mathbf{0}$, $\mathrm{Cov}(\boldsymbol{\tau}) = \sigma_\tau^2 \mathbf{R}$, and **R** is a matrix that determines the spatial dependence structure in **y** and depends on a range parameter, $\varphi$. We discuss **R** in more detail shortly. The parameter $\sigma_\tau^2$ is called the spatially dependent random error variance or partial sill. The parameter $\sigma_\epsilon^2$ is called the spatially independent random error variance or nugget. These two variance parameters are henceforth more intuitively written as $\sigma_{de}^2$ and $\sigma_{ie}^2$, respectively. The covariance of **y** is denoted $\Sigma$ and given by $\sigma_{de}^2 \mathbf{R} + \sigma_{ie}^2 \mathbf{I}$. The parameters that compose this covariance are contained in the vector $\boldsymbol{\theta}$, which is called the covariance parameter vector.

Eq 2 is called the spatial linear model. The spatial linear model applies to both point-referenced and areal data. The `splm()` function is used to fit spatial linear models for point-referenced data (i.e., geostatistical models). One spatial covariance function available in `splm()` is the exponential spatial covariance function, which has an **R** matrix given by

$$\mathbf{R} = \exp(-\mathbf{M}/\phi), \tag{3}$$

where **M** is a matrix of Euclidean distances among observations. Recall that $\varphi$ is the range parameter, and it controls the behavior of **R** as a function of distance. In Eq 3, as the distance between two observations increases, the correlation between them decreases. Parameterizations for other `splm()` spatial covariance types and their **R** matrices can be viewed by running `help("splm", "spmodel")` or `vignette("technical", "spmodel")`. Some of these spatial covariance types (e.g., Matérn) depend on an extra parameter beyond $\sigma_{de}^2$, $\sigma_{ie}^2$, and $\phi$.

The `spautor()` function is used to fit spatial linear models for areal data (i.e., spatial autoregressive models). One spatial autoregressive covariance function available in `spautor()` is the simultaneous autoregressive spatial covariance function, which has an **R** matrix given by

$$\mathbf{R} = [(\mathbf{I} - \phi\mathbf{W})(\mathbf{I} - \phi\mathbf{W})^\top]^{-1},$$

where **W** is a weight matrix describing the neighborhood structure in **y**. Parameterizations for `spautor()` spatial covariance types and their **R** matrices can be seen by running `help("spautor", "spmodel")` or `vignette("technical", "spmodel")`.

One way to define **W** is through queen contiguity [23]. Two observations are queen contiguous if they share a boundary. The $ij$th element of **W** is then one if observation $i$ and observation $j$ are queen contiguous and zero otherwise. Observations are not considered neighbors with themselves, so each diagonal element of **W** is zero.

Sometimes each element in the weight matrix **W** is divided by its respective row sum. This is called row-standardization. Row-standardizing **W** has several benefits, which are discussed in detail by [24].

## Model fitting

In this section, we show how to use the `splm()` and `spautor()` functions to estimate parameters of the spatial linear model. We also explore diagnostic tools in `spmodel` that evaluate model fit. The `splm()` and `spautor()` functions share similar syntactic structure with the `lm()` function used to fit non-spatial linear models from Eq 1. `splm()` and `spautor()` generally require at least three arguments:

- `formula`: a formula that describes the relationship between the response variable (**y**) and explanatory variables (**X**)

– `formula` in `splm()` is the same as `formula` in `lm()`

- `data`: a `data.frame` or `sf` object that contains the response variable, explanatory variables, and spatial information

- `spcov_type`: the spatial covariance type ("`exponential`", "`matern`", "`car`", etc)

If `data` is an `sf` [25] object, spatial information is stored in the object's geometry. If `data` is a `data.frame`, then the x-coordinates and y-coordinates must be provided via the `xcoord` and `ycoord` arguments (for point-referenced data) or the weight matrix must be provided via the `W` argument (for areal data).

In the following subsections, we use the point-referenced `moss` data, an `sf` object that contains data on heavy metals in mosses near a mining road in Alaska. We view the first few rows of `moss` by running

```
R> moss

Simple feature collection with 365 features and 7 fields
Geometry type: POINT
Dimension:     XY
Bounding box:  xmin: -445884.1 ymin: 1929616 xmax: -383656.8 ymax: 2061414
Projected CRS: NAD83 / Alaska Albers
# A tibble: 365 x 8
   sample field_dup lab_rep year  sideroad log_dist2road log_Zn
   <fct>  <fct>     <fct>   <fct> <fct>            <dbl>  <dbl>
 1 001PR  1         1       2001  N                 2.68   7.33
 2 001PR  1         2       2001  N                 2.68   7.38
 3 002PR  1         1       2001  N                 2.54   7.58
 4 003PR  1         1       2001  N                 2.97   7.63
 5 004PR  1         1       2001  N                 2.72   7.26
 6 005PR  1         1       2001  N                 2.76   7.65
 7 006PR  1         1       2001  S                 2.30   7.59
 8 007PR  1         1       2001  N                 2.78   7.16
 9 008PR  1         1       2001  N                 2.93   7.19
10 009PR  1         1       2001  N                 2.79   8.07
# ... with 355 more rows, and 1 more variable:
#   geometry <POINT [m]>
```

We can learn more about `moss` by running `help("moss", "spmodel")`, and we can visualize the distribution of log zinc concentration in `moss` (Fig 1) by running

```
R> ggplot(moss, aes(color = log_Zn)) +
+   geom_sf(size = 2) +
+   scale_color_viridis_c() +
+   theme_gray(base_size = 14)
```

Log zinc concentration can be viewed interactively in `mapview` by running

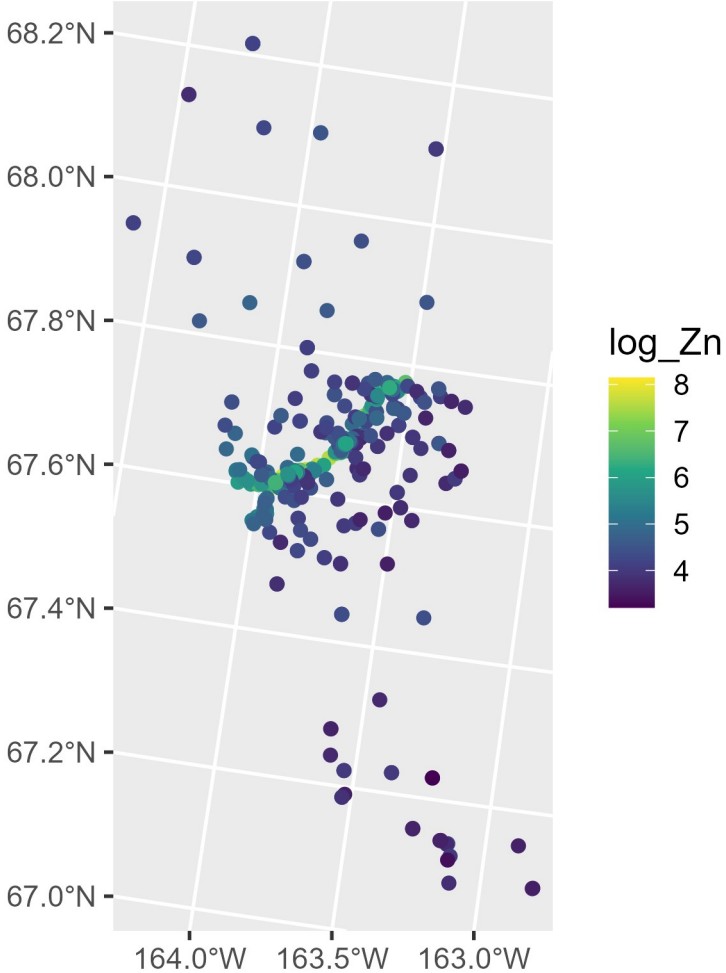

**Fig 1. Distribution of log zinc concentration in the moss data.**

```
R> mapview(moss, zcol = "log_Zn")
```

## Estimation

Generally the covariance parameters ($\theta$) and fixed effects ($\beta$) of the spatial linear model require estimation. The default estimation method in spmodel is restricted maximum likelihood [26–28]. Maximum likelihood estimation is also available. For point-referenced data, semivariogram weighted least squares [29] and semivariogram composite likelihood [30] are additional estimation methods. The estimation method is chosen using the estmethod argument.

We estimate parameters of a spatial linear model regressing log zinc concentration (log_Zn) on log distance to a haul road (log_dist2road) using an exponential spatial covariance function by running

```
R> spmod <- splm(log_Zn ~ log_dist2road, moss, spcov_type = "exponen-
tial")
```

We summarize the model fit by running

```
R> summary(spmod)
```

```
Call:
splm(formula = log_Zn ~ log_dist2road, data = moss, spcov_type = "exponen-
tial")
```

```
Residuals:
    Min      1Q   Median      3Q      Max
-2.6801 -1.3606 -0.8103 -0.2485  1.1298
```

```
Coefficients (fixed):
              Estimate Std. Error z value Pr(>|z|)
(Intercept)    9.76825    0.25216   38.74   <2e-16 ***
log_dist2road -0.56287    0.02013  -27.96   <2e-16 ***
---
Signif. codes:  0 '***' 0.001 '**' 0.01 '*' 0.05 '.' 0.1 ' ' 1
```

```
Pseudo R-squared: 0.683
```

```
Coefficients (exponential spatial covariance):
      de        ie      range
3.595e-01 7.897e-02 8.237e+03
```

The fixed effects coefficient table contains estimates, standard errors, z-statistics, and asymptotic p-values for each fixed effect. From this table, we notice there is evidence that mean log zinc concentration significantly decreases with distance from the haul road (p-value < 2e-16). We see the fixed effect estimates by running

```
R> coef(spmod)
```

```
  (Intercept) log_dist2road
    9.7682525    -0.5628713
```

The model summary also contains the exponential spatial covariance parameter estimates, which we can view by running

```
R> coef(spmod, type = "spcov")
          de           ie        range         rotate        scale
3.595316e-01 7.896824e-02 8.236712e+03 0.000000e+00 1.000000e+00
attr(,"class")
[1] "exponential"
```

The dependent random error variance ($\sigma^2_{de}$) is estimated to be approximately 0.36 and the independent random error variance ($\sigma^2_{ie}$) is estimated to be approximately 0.079. The range ($\phi$) is estimated to be approximately 8,237. The effective range is the distance at which the spatial covariance is approximately zero. For the exponential covariance, the effective range is $3\phi$. This means that observations whose distance is greater than 24,711 meters are approximately uncorrelated. The `rotate` and `scale` parameters affect the modeling of anisotropy, which we discuss later. By default, `rotate` and `scale` are assumed to be zero and one, respectively,

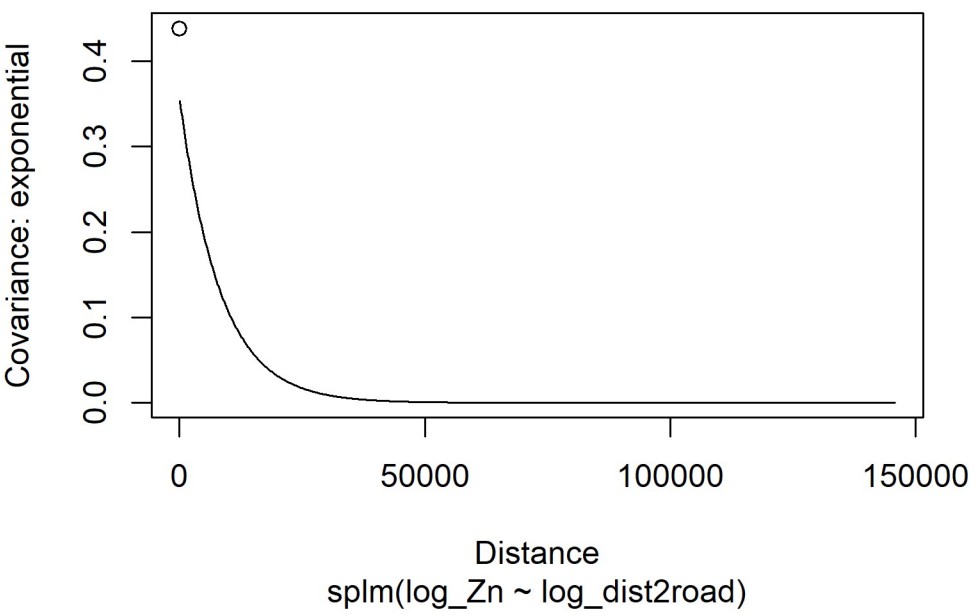

**Fig 2. Empirical spatial covariance of the fitted model.** The open circle at a distance of zero represents the $\sigma^2_{de} + \sigma^2_{ie}$. The solid line at positive distances represents $\sigma^2_{de}\mathbf{R}$ at a particular distance.

which means that anisotropy is not modeled (i.e., the spatial covariance is assumed isotropic, or independent of direction). We visualize the fitted spatial covariance function (Fig 2) by running

```
R> plot(spmod, which = 7)
```

## Model-fit statistics

The quality of model fit can be assessed using a variety of statistics readily available in `spmodel`. The first model-fit statistic we consider is the pseudo R-squared. The pseudo R-squared is a generalization of the classical R-squared from non-spatial linear models that quantifies the proportion of variability in the data explained by the fixed effects. The pseudo R-squared is defined as

$$PR2 = 1 - \frac{\mathcal{D}_{\hat{\boldsymbol{\beta}}}}{\mathcal{D}_{\hat{\mu}}},$$

where $\mathcal{D}_{\hat{\boldsymbol{\beta}}}$ is the deviance of the fitted model with all explanatory variables and $\mathcal{D}_{\hat{\mu}}$ is the deviance of the fitted model with only an intercept. We compute the pseudo R-squared by running

```
R> pseudoR2(spmod)
[1] 0.6829687
```

Roughly 68% of the variability in log zinc is explained by log distance from the road. The pseudo R-squared can be adjusted to account for the number of explanatory variables using the `adjust` argument. Pseudo R-squared (and the adjusted version) is most helpful for comparing models that have the same covariance structure.

The next two model-fit statistics we consider are the AIC and AICc that [31] derive for spatial data. The AIC and AICc evaluate the fit of a model with a penalty for the number of parameters estimated. This penalty balances model fit and model parsimony. Lower AIC and AICc indicate a better balance of model fit and parsimony. The AICc is a correction to AIC that is better suited for small sample sizes. As the sample size increases, AIC and AICc converge.

The AIC and AICc are given by

$$
\begin{aligned}
\text{AIC} \quad &= -2\ell(\hat{\boldsymbol{\Theta}}) + 2(|\hat{\boldsymbol{\Theta}}|) \\
\text{AICc} \quad &= -2\ell(\hat{\boldsymbol{\Theta}}) + 2n(|\hat{\boldsymbol{\Theta}}|)/(n - |\hat{\boldsymbol{\Theta}}| - 1),
\end{aligned}
$$

where $\ell(\hat{\boldsymbol{\Theta}})$ is the log-likelihood of the data evaluated at the estimated parameter vector $\hat{\boldsymbol{\Theta}}$ that maximized $\ell(\boldsymbol{\Theta})$, $|\hat{\boldsymbol{\Theta}}|$ is the cardinality of $\hat{\boldsymbol{\Theta}}$, and $n$ is the sample size. For maximum likelihood, $\hat{\boldsymbol{\Theta}} = \{\hat{\boldsymbol{\Theta}}, \hat{\boldsymbol{\beta}}\}$, and for restricted maximum likelihood, $\hat{\boldsymbol{\Theta}} = \{\hat{\boldsymbol{\Theta}}\}$. There are some nuances to consider when comparing AIC across models: AIC comparisons between a model fit using restricted maximum likelihood and a model fit using maximum likelihood are meaningless, as the models are fit with different likelihoods; and AIC comparisons between models fit using restricted maximum likelihood are only valid when the models have the same fixed effect structure; AIC comparisons between models fit using maximum likelihood are valid even when the models have different fixed effect structures [32].

Suppose we want to quantify the difference in model quality between the spatial model and a non-spatial model using the AIC and AICc criteria. We fit a non-spatial model (Eq 1) in `spmodel` by running

```
R> lmod <- splm(log_Zn ~ log_dist2road, moss, spcov_type = "none")
```

This model is equivalent to one fit using `lm()`. We compute the spatial AIC and AICc of the spatial model and non-spatial model by running

```
R> AIC(spmod, lmod)

        df      AIC
spmod   3  373.2089
lmod    1  636.0635
```

```
R> AICc(spmod, lmod)

        df     AICc
spmod   3  373.2754
lmod    1  636.0745
```

The noticeably lower AIC and AICc of of the spatial model indicate that it is a better fit to the data than the non-spatial model. Recall that these AIC and AICc comparisons are valid because both models are fit using restricted maximum likelihood (the default).

Another approach to comparing the fitted models is to perform leave-one-out cross validation [33]. In leave-one-out cross validation, a single observation is removed from the data, the model is re-fit, and a prediction is made for the held-out observation. Then, a loss metric like mean-squared-prediction error is computed and used to evaluate model fit. The lower the mean-squared-prediction error, the better the model fit. For computational efficiency, leave-one-out cross validation in `spmodel` is performed by first estimating $\boldsymbol{\theta}$ using all the data and

then re-estimating $\boldsymbol{\beta}$ for each observation. We perform leave-one-out cross validation for the spatial and non-spatial model by running

```
R> loocv(spmod)
```

```
[1] 0.1110895
```

```
R> loocv(lmod)
```

```
[1] 0.3237897
```

The noticeably lower mean-squared-prediction error of the spatial model indicates that it is a better fit to the data than the non-spatial model.

## Diagnostics

In addition to model fit metrics, spmodel provides functions to compute diagnostic metrics that help assess model assumptions and identify unusual observations.

An observation is said to have high leverage if its combination of explanatory variable values is far from the mean vector of the explanatory variables. For a non-spatial model, the leverage of the $i$th observation is the $i$th diagonal element of the hat matrix given by

$$\mathbf{H} = \mathbf{X}(\mathbf{X}^\top \mathbf{X})^{-1} \mathbf{X}^\top.$$

For a spatial model, the leverage of the $i$th observation is the $i$th diagonal element of the spatial hat matrix given by

$$\mathbf{H}^* = (\mathbf{X}^*(\mathbf{X}^{*\top}\mathbf{X})^{-1}\mathbf{X}^{*\top}),$$

where $\mathbf{X}^* = \Sigma^{-1/2}\mathbf{X}$ and $\Sigma^{-1/2}$ is the inverse square root of the covariance matrix, $\Sigma$ [34]. The spatial hat matrix can be viewed as the non-spatial hat matrix applied to $\mathbf{X}^*$ instead of $\mathbf{X}$. We compute the hat values (leverage) by running

```
R> hatvalues(spmod)
```

Larger hat values indicate more leverage, and observations with large hat values may be unusual and warrant further investigation.

The fitted value of an observation is the estimated mean response given the observation's explanatory variable values and the model fit:

$$\hat{\mathbf{y}} = \mathbf{X}\hat{\boldsymbol{\beta}}.$$

We compute the fitted values by running

```
R> fitted(spmod)
```

Fitted values for the spatially dependent random errors ($\boldsymbol{\tau}$), spatially independent random errors ($\epsilon$), and random effects can also be obtained via fitted() by changing the type argument.

The residuals measure each response's deviation from its fitted value. The response residuals are given by

$$\mathbf{e}_r = \mathbf{y} - \hat{\mathbf{y}}.$$

We compute the response residuals of the spatial model by running

```
R> residuals(spmod)
```

The response residuals are typically not directly checked for linear model assumptions, as they have covariance closely resembling the covariance of **y**. Pre-multiplying the residuals by $\boldsymbol{\Sigma}^{-1/2}$ yields the Pearson residuals [35]:

$$\mathbf{e}_p = \boldsymbol{\Sigma}^{-1/2}\mathbf{e}_r.$$

When the model is correct, the Pearson residuals have mean zero, variance approximately one, and are uncorrelated. We compute the Pearson residuals of the spatial model by running

```
R> residuals(spmod, type = "pearson")
```

The covariance of $\mathbf{e}_p$ is $(\mathbf{I} - \mathbf{H}^*)$, which is approximately **I** for large sample sizes. Explicitly dividing $\mathbf{e}_p$ by the respective diagonal element of $(\mathbf{I} - \mathbf{H}^*)$ yields the standardized residuals [35]:

$$\mathbf{e}_s = \frac{\mathbf{e}_p}{\sqrt{(1 - \text{diag}(\mathbf{H}^*))}},$$

where $\text{diag}(\mathbf{H}^*)$ denotes the diagonal of $\mathbf{H}^*$. We compute the standardized residuals of the spatial model by running

```
R> residuals(spmod, type = "standardized")
```

or

```
R> rstandard(spmod)
```

When the model is correct, the standardized residuals have mean zero, variance one, and are uncorrelated.

It is common to check linear model assumptions through visualizations. We can visualize the standardized residuals vs fitted values by running

```
R> plot(spmod, which = 1) # figure omitted
```

When the model is correct, the standardized residuals should be evenly spread around zero with no discernible pattern. We can visualize a normal QQ-plot of the standardized residuals by running

```
R> plot(spmod, which = 2) # figure omitted
```

When the standardized residuals are normally distributed, they should closely follow the normal QQ-line.

An observation is said to be influential if its omission has a large impact on model fit. Typically, this is measured using Cook's distance [36]. For the non-spatial model, the Cook's distance of the $i$th observation is denoted **D** and given by

$$\mathbf{D} = \mathbf{e}_s^2 \frac{\text{diag}(\mathbf{H})}{p(1 - \text{diag}(\mathbf{H}))},$$

where $p$ is the dimension of $\boldsymbol{\beta}$ (the number of fixed effects).

For a spatial model, the Cook's distance of the $i$th observation is denoted $\mathbf{D}^*$ and given by

$$\mathbf{D}^* = \mathbf{e}_s^2 \frac{\text{diag}(\mathbf{H}^*)}{p(1 - \text{diag}(\mathbf{H}^*))}.$$

A larger Cook's distance indicates more influence, and observations with large Cook's distance values may be unusual and warrant further investigation. We compute Cook's distance by running

```
R> cooks.distance(spmod)
```

The Cook's distance versus leverage (hat values) can be visualized by running

```
R> plot(spmod, which = 6) # figure omitted
```

Though we described the model diagnostics in this subsection using $\Sigma$, generally the covariance parameters are estimated and $\Sigma$ is replaced with $\hat{\Sigma}$.

### The broom functions: `tidy()`, `glance()`, and `augment()`

The `tidy()`, `glance()`, and `augment()` functions from the broom **R** package [37] provide convenient output for many of the model fit and diagnostic metrics discussed in the previous two sections. The `tidy()` function returns a tidy tibble of the coefficient table from `summary()`:

```
R> tidy(spmod)

# A tibble: 2 x 5
  term          estimate std.error statistic p.value
  <chr>            <dbl>     <dbl>     <dbl>   <dbl>
1 (Intercept)       9.77    0.252      38.7       0
2 log_dist2road    -0.563   0.0201    -28.0       0
```

This tibble format makes it easy to pull out the coefficient names, estimates, standard errors, z-statistics, and p-values from the `summary()` output.

The `glance()` function returns a tidy tibble of model-fit statistics:

```
R> glance(spmod)

# A tibble: 1 x 9
      n     p npar value   AIC  AICc logLik deviance pseudo.r.squared
  <int> <dbl> <int> <dbl> <dbl> <dbl>  <dbl>    <dbl>            <dbl>
1   365     2     3  367.  373.  373.  -184.      363            0.683
```

The `glances()` function is an extension of `glance()` that can be used to look at many models simultaneously:

```
R> glances(spmod, lmod)

# A tibble: 2 x 10
  model     n     p npar value   AIC  AICc logLik deviance
  <chr> <int> <dbl> <int> <dbl> <dbl> <dbl>  <dbl>    <dbl>
1 spmod   365     2     3  367.  373.  373.  -184.      363
2 lmod    365     2     1  634.  636.  636.  -317.      363.
# ... with 1 more variable: pseudo.r.squared <dbl>
```

Finally, the `augment()` function augments the original data with model diagnostics:

```
R> augment(spmod)
```

```
Simple feature collection with 365 features and 7 fields
Geometry type: POINT
Dimension:     XY
Bounding box:  xmin: −445884.1 ymin: 1929616 xmax: −383656.8 ymax: 2061414
Projected CRS: NAD83 / Alaska Albers
# A tibble: 365 x 8
   log_Zn log_dist2road .fitted .resid    .hat  .cooksd .std.resid
 * <dbl>          <dbl>   <dbl>  <dbl>   <dbl>    <dbl>      <dbl>
 1   7.33           2.68    8.26 −0.928 0.102    0.112      −1.48
 2   7.38           2.68    8.26 −0.880 0.0101   0.000507   −0.316
 3   7.58           2.54    8.34 −0.755 0.0170   0.000475   −0.236
 4   7.63           2.97    8.09 −0.464 0.0137   0.000219    0.178
 5   7.26           2.72    8.24 −0.977 0.0177   0.00515    −0.762
 6   7.65           2.76    8.21 −0.568 0.0147   0.000929   −0.355
 7   7.59           2.30    8.47 −0.886 0.0170   0.00802    −0.971
 8   7.16           2.78    8.20 −1.05  0.0593   0.0492     −1.29
 9   7.19           2.93    8.12 −0.926 0.00793  0.000451   −0.337
10   8.07           2.79    8.20 −0.123 0.0265   0.00396     0.547
# ... with 355 more rows, and 1 more variable: geometry <POINT [m]>
```

By default, only the columns of `data` used to fit the model are returned alongside the diagnostics. All original columns of `data` are returned by setting `drop` to `FALSE`. `augment()` is especially powerful when the data are an `sf` object because model diagnostics can be easily visualized spatially. For example, we could subset the augmented object so that it only includes observations whose standardized residuals have absolute values greater than some cutoff and then map them.

### An areal data example

Next we use the `seal` data, an `sf` object that contains the log of the estimated harbor-seal trends from abundance data across polygons in Alaska, to provide an example of fitting a spatial linear model for areal data using `spautor()`. We view the first few rows of `seal` by running

```
R> seal
```

```
Simple feature collection with 62 features and 1 field
Geometry type: POLYGON
Dimension:     XY
Bounding box:  xmin: 913618.8 ymin: 1007542 xmax: 1116002 ymax: 1145054
Projected CRS: NAD83 / Alaska Albers
# A tibble: 62 x 2
   log_trend              geometry
       <dbl>         <POLYGON [m]>
 1 NA       ((1035002 1054710, 1035002~
 2 −0.282   ((1037002 1039492, 1037006~
 3 −0.00121 ((1070158 1030216, 1070185~
 4  0.0354  ((1054906 1034826, 1054931~
 5 −0.0160  ((1025142 1056940, 1025184~
 6  0.0872  ((1026035 1044623, 1026037~
```

```
 7  -0.266   ((1100345 1060709, 1100287~
 8   0.0743  ((1030247 1029637, 1030248~
 9  NA       ((1043093 1020553, 1043097~
10  -0.00961 ((1116002 1024542, 1116002~
# ... with 52 more rows
```

We can learn more about the data by running `help("seal", "spmodel")`.

We can visualize the distribution of log seal trends in the `seal` data (Fig 3) by running

```
R> ggplot(seal, aes(fill = log_trend)) +
+   geom_sf(size = 0.75) +
+   scale_fill_viridis_c() +
+   theme_bw(base_size = 14)
```

Log trends can be viewed interactively in `mapview` by running

```
R> mapview(seal, zcol = "log_trend")
```

The gray polygons denote areas where the log trend is missing. These missing areas need to be kept in the data while fitting the model to preserve the overall neighborhood structure.

We estimate parameters of a spatial autoregressive model for log seal trends (`log_trend`) using an intercept-only model with a conditional autoregressive (CAR) spatial covariance by running

```
R> sealmod <- spautor(log_trend ~ 1, seal, spcov_type = "car")
```

If a weight matrix is not provided to `spautor()`, it is calculated internally using queen contiguity. Recall that queen contiguity defines two observations as neighbors if they share at least one common boundary. If at least one observation has no neighbors, the `extra` parameter is estimated, which quantifies variability among observations without neighbors. By default, `spautor()` uses row standardization [24] and assumes an independent error variance (`ie`) of zero.

We summarize, tidy, glance at, and augment the fitted model by running

```
R> summary(sealmod)
```

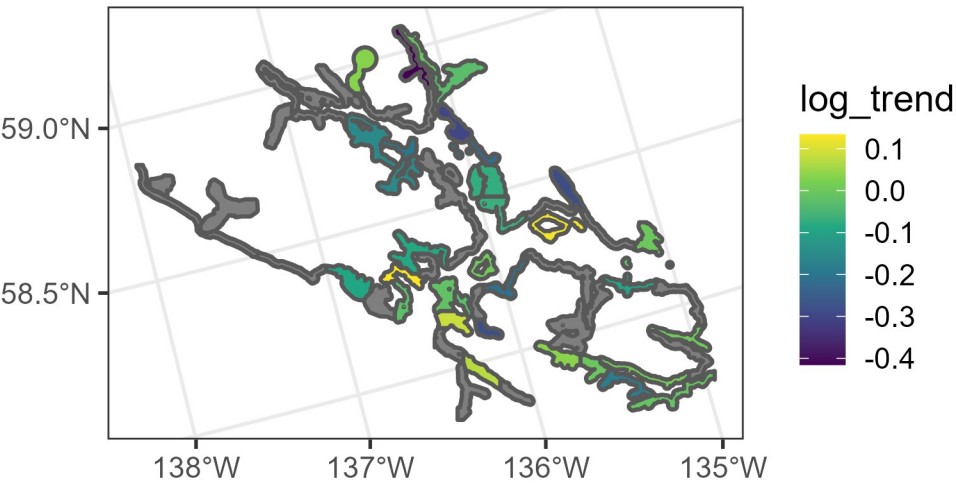

**Fig 3. Distribution of log seal trends in the seal data.** Polygons are gray if seal trends are missing.

```
Call:
spautor(formula = log_trend ~ 1, data = seal, spcov_type = "car")

Residuals:
    Min      1Q   Median      3Q      Max
-0.34443 -0.10405  0.04422  0.07349  0.20487

Coefficients (fixed):
            Estimate Std. Error z value Pr(>|z|)
(Intercept) -0.07102    0.02495  -2.846  0.00443 **
---
Signif. codes:
0 '***' 0.001 '**' 0.01 '*' 0.05 '.' 0.1 ' ' 1

Coefficients (car spatial covariance):
     de   range   extra
0.03261 0.41439 0.02221

R> tidy(sealmod)

# A tibble: 1 x 5
  term        estimate std.error statistic p.value
  <chr>          <dbl>     <dbl>     <dbl>   <dbl>
1 (Intercept)  -0.0710    0.0250     -2.85 0.00443

R> glance(sealmod)

# A tibble: 1 x 9
      n     p  npar value   AIC   AICc logLik deviance pseudo.r.squared
  <int> <dbl> <int> <dbl> <dbl>  <dbl>  <dbl>    <dbl>            <dbl>
1    34     1     3 -36.9 -30.9  -30.1   18.4     32.9                0

R> augment(sealmod)

Simple feature collection with 34 features and 6 fields
Geometry type: POLYGON
Dimension:      XY
Bounding box:  xmin: 980001.5 ymin: 1010815 xmax: 1116002 ymax: 1145054
Projected CRS: NAD83 / Alaska Albers
# A tibble: 34 x 7
   log_trend .fitted  .resid   .hat .cooksd .std.resid
 *     <dbl>   <dbl>   <dbl>  <dbl>   <dbl>      <dbl>
 1  -0.282   -0.0710 -0.211  0.0179 0.0233      -1.14
 2  -0.00121 -0.0710  0.0698 0.0699 0.0412       0.767
 3   0.0354  -0.0710  0.106  0.0218 0.0109       0.705
 4  -0.0160  -0.0710  0.0550 0.0343 0.00633      0.430
 5   0.0872  -0.0710  0.158  0.0229 0.0299       1.14
 6  -0.266   -0.0710 -0.195  0.0280 0.0493      -1.33
 7   0.0743  -0.0710  0.145  0.0480 0.0818       1.30
```

```
 8  -0.00961 -0.0710  0.0614 0.0143 0.00123      0.293
 9  -0.182   -0.0710 -0.111  0.0131 0.0155      -1.09
10   0.00351 -0.0710  0.0745 0.0340 0.0107       0.561
# ... with 24 more rows, and 1 more variable:
#   geometry <POLYGON [m]>
```

Note that for `spautor()` models, the `ie` spatial covariance parameter is assumed zero by default (and omitted from the `summary()` output). This default behavior can be overridden by specifying `ie` in the `spcov_initial` argument to `spautor()`. Also note that the pseudo R-squared is zero because there are no explanatory variables in the model (i.e., it is an intercept-only model).

## Prediction

In this section, we show how to use `predict()` to perform spatial prediction (also called Kriging) in `spmodel`. We will fit a model using the point-referenced `sulfate` data, an `sf` object that contains sulfate measurements in the conterminous United States, and make predictions for each location in the point-referenced `sulfate_preds` data, an `sf` object that contains locations in the conterminous United States at which to predict sulfate.

We first visualize the distribution of the sulfate data (Fig 4A) by running

```
R> ggplot(sulfate, aes(color = sulfate)) +
+   geom_sf(size = 2.5) +
+   scale_color_viridis_c(limits = c(0, 45)) +
+   theme_gray(base_size = 18)
```

We then fit a spatial linear model for sulfate using an intercept-only model with a spherical spatial covariance function by running

```
R> sulfmod <- splm(sulfate ~ 1, sulfate, spcov_type = "spherical")
```

Then we obtain best linear unbiased predictions (Kriging predictions) using `predict()`. The `newdata` argument contains the locations at which to predict, and we store the predictions as a new variable in `sulfate_preds` called `preds` by running

```
R> sulfate_preds$preds <- predict(sulfmod, newdata = sulfate_preds)
```

We can visualize the model predictions (Fig 4B) by running

```
R> ggplot(sulfate_preds, aes(color = preds)) +
+   geom_sf(size = 2.5) +
+   scale_color_viridis_c(limits = c(0, 45)) +
+   theme_gray(base_size = 18)
```

It is important to properly specify the `newdata` object when running `predict()`. If explanatory variables were used to fit the model, the same explanatory variables must be included in `newdata` with the same names as they have in `data`. Additionally, if an explanatory variable is categorical or a factor, the values of this variable in `newdata` must also be values in `data` (e.g., if a categorical variable with values "A", and "B" was used to fit the model, the corresponding variable in `newdata` cannot have a value "C"). If `data` is a `data.`

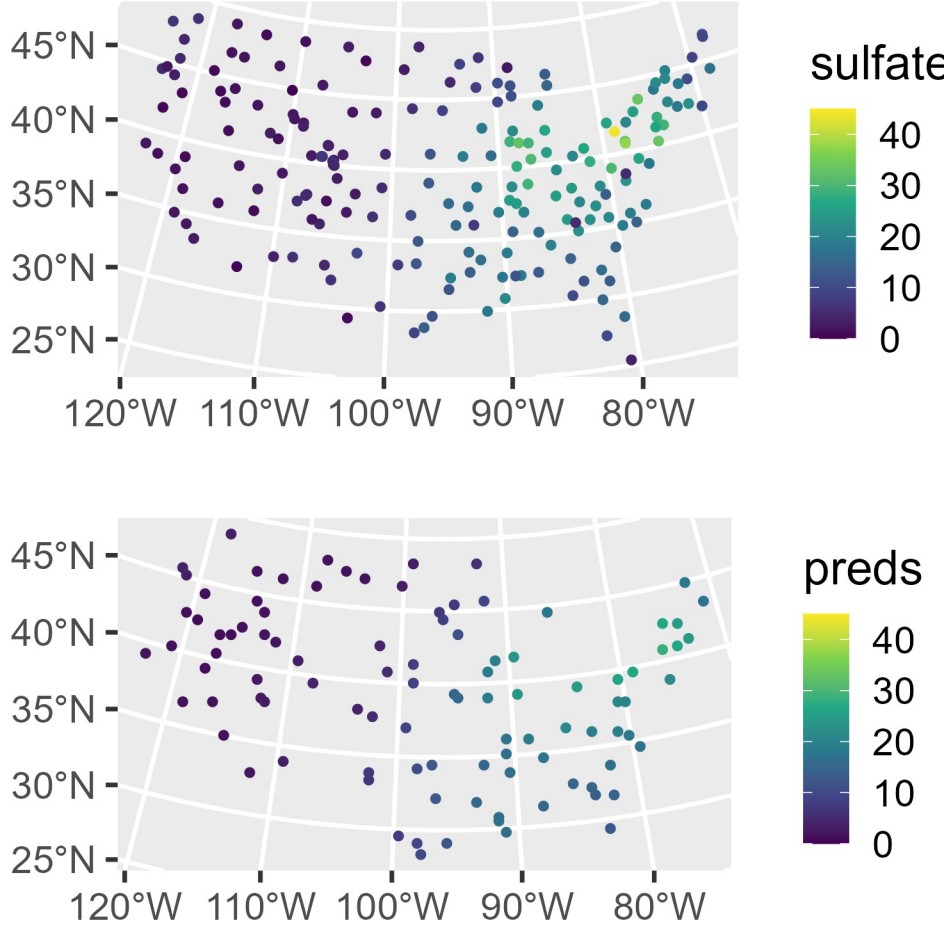

**Fig 4. Distribution of observed sulfate and sulfate predictions in the conterminous United States.** In A (top), observed sulfate is visualized. In B (bottom), sulfate predictions are visualized.

frame, coordinates must be included in newdata with the same names as they have in data. If data is an sf object, coordinates must be included in newdata with the same geometry name as they have in data. When using projected coordinates, the projection for newdata should be the same as the projection for data.

Prediction standard errors are returned by setting the se.fit argument to TRUE:

```
R> predict(sulfmod, newdata = sulfate_preds, se.fit = TRUE)
```

The interval argument determines the type of interval returned. If interval is "none" (the default), no intervals are returned. If interval is "prediction", then 100 * level% prediction intervals are returned (the default is 95% prediction intervals):

```
R> predict(sulfmod, newdata = sulfate_preds, interval = "prediction")
```

If interval is "confidence", the predictions are instead the estimated mean given each observation's explanatory variable values (i.e., fitted values) and the corresponding 100 * level% confidence intervals are returned:

```
R> predict(sulfmod, newdata = sulfate_preds, interval = "confidence")
```

The `predict()` output structure changes based on `interval` and `se.fit`. For more details, run `help("predict.spmodel", "spmodel")`.

Previously we used the `augment()` function to augment `data` with model diagnostics. We can also use `augment()` to augment `newdata` with predictions, standard errors, and intervals. We remove the model predictions from `sulfate_preds` before showing how `augment()` is used to obtain the same predictions by running

```
R> sulfate_preds$preds <- NULL
```

We then view the first few rows of `sulfate_preds` augmented with a 90% prediction interval by running

```
R> augment(
+   sulfmod,
+   newdata = sulfate_preds,
+   interval = "prediction",
+   level = 0.90
+ )
Simple feature collection with 100 features and 3 fields
Geometry type: POINT
Dimension:     XY
Bounding box:  xmin: -2283774 ymin: 582930.5 xmax: 1985906 ymax: 3037173
Projected CRS: NAD83 / Conus Albers
# A tibble: 100 x 4
   .fitted .lower .upper           geometry
 *   <dbl>  <dbl>  <dbl>        <POINT [m]>
 1    1.40  -5.33   8.14 (-1771413 1752976)
 2   24.5   18.2   30.8    (1018112 1867127)
 3    8.99   2.36  15.6   (-291256.8 1553212)
 4   16.4    9.92  23.0    (1274293 1267835)
 5    4.91  -1.56  11.4   (-547437.6 1638825)
 6   26.7   20.4   33.0    (1445080 1981278)
 7    3.00  -3.65   9.66  (-1629090 3037173)
 8   14.3    7.97  20.6    (1302757 1039534)
 9    1.49  -5.08   8.06  (-1429838 2523494)
10   14.4    7.97  20.8    (1131970 1096609)
# ... with 90 more rows
```

Here `.fitted` represents the predictions, `.lower` represents the lower bound of the 90% prediction intervals, and `.upper` represents the upper bound of the 90% prediction intervals.

An alternative (but equivalent) approach can be used for model fitting and prediction that circumvents the need to keep `data` and `newdata` as separate objects. Suppose that observations requiring prediction are stored in `data` as missing (`NA`) values. We can add a column of missing values to `sulfate_preds` and then bind it together with `sulfate` by running

```
R> sulfate_preds$sulfate <- NA
R> sulfate_with_NA <- rbind(sulfate, sulfate_preds)
```

We can then fit a spatial linear model by running

```
R> sulfmod_with_NA <- splm(
+   sulfate ~ 1,
```

```
+    sulfate_with_NA,
+    spcov_type = "spherical"
+ )
```

The missing values are ignored for model-fitting but stored in `sulfmod_with_NA` as `newdata`:

```
R> sulfmod_with_NA$newdata
```

```
Simple feature collection with 100 features and 1 field
Geometry type: POINT
Dimension:      XY
Bounding box:   xmin: −2283774 ymin: 582930.5 xmax: 1985906 ymax: 3037173
Projected CRS: NAD83 / Conus Albers
First 10 features:
    sulfate                geometry
198     NA  POINT (−1771413 1752976)
199     NA   POINT (1018112 1867127)
200     NA POINT (−291256.8 1553212)
201     NA   POINT (1274293 1267835)
202     NA POINT (−547437.6 1638825)
203     NA   POINT (1445080 1981278)
204     NA  POINT (−1629090 3037173)
205     NA   POINT (1302757 1039534)
206     NA  POINT (−1429838 2523494)
207     NA   POINT (1131970 1096609)
```

We can then predict the missing values by running

```
R> predict(sulfmod_with_NA)
```

The call to `predict()` finds in `sulfmod_with_NA` the `newdata` object and is equivalent to

```
R> predict(sulfmod_with_NA, newdata = sulfmod_with_NA$newdata)
```

We can also use `augment()` to make the predictions for the data set with missing values by running

```
R> augment(sulfmod_with_NA, newdata = sulfmod_with_NA$newdata)
```

```
Simple feature collection with 100 features and 2 fields
Geometry type: POINT
Dimension:      XY
Bounding box:   xmin: −2283774 ymin: 582930.5 xmax: 1985906 ymax: 3037173
Projected CRS: NAD83 / Conus Albers
# A tibble: 100 x 3
   sulfate .fitted          geometry
 *   <dbl>   <dbl>       <POINT [m]>
 1      NA    1.40  (−1771413 1752976)
 2      NA   24.5    (1018112 1867127)
 3      NA    8.99 (−291256.8 1553212)
 4      NA   16.4    (1274293 1267835)
```

```
 5      NA   4.91 (-547437.6 1638825)
 6      NA   26.7   (1445080 1981278)
 7      NA    3.00 (-1629090 3037173)
 8      NA   14.3    (1302757 1039534)
 9      NA    1.49 (-1429838 2523494)
10      NA   14.4    (1131970 1096609)
# ... with 90 more rows
```

Unlike `predict()`, `augment()` explicitly requires the `newdata` argument be specified in order to obtain predictions. Omitting `newdata` (e.g., running `augment (sulfmod_with_NA)`) returns model diagnostics, not predictions.

For areal data models fit with `spautor()`, predictions cannot be computed at locations that were not incorporated in the neighborhood structure used to fit the model. Thus, predictions are only possible for observations in `data` whose response values are missing (`NA`), as their locations are incorporated into the neighborhood structure. For example, we make predictions of log seal trends at the missing polygons from Fig 3 by running

```
R> predict(sealmod)
```

We can also use `augment()` to make the predictions:

```
R> augment(sealmod, newdata = sealmod$newdata)

Simple feature collection with 28 features and 2 fields
Geometry type: POLYGON
Dimension:      XY
Bounding box:  xmin: 913618.8 ymin: 1007542 xmax: 1115097 ymax: 1132682
Projected CRS: NAD83 / Alaska Albers
# A tibble: 28 x 3
   log_trend .fitted
 *     <dbl>   <dbl>
 1        NA -0.113
 2        NA -0.0108
 3        NA -0.0608
 4        NA -0.0383
 5        NA -0.0730
 6        NA -0.0556
 7        NA -0.0968
 8        NA -0.0716
 9        NA -0.0776
10        NA -0.0647
# ... with 18 more rows, and 1 more
#   variable: geometry <POLYGON [m]>
```

## Advanced features

`spmodel` offers several advanced features for fitting spatial linear models. We briefly discuss some of these features next using the `moss` data and some simulated data. Technical details for each advanced feature can be seen by running `vignette("technical", "spmodel")`.

## Fixing spatial covariance parameters

We may desire to fix specific spatial covariance parameters at a particular value. Perhaps some parameter value is known, for example. Or perhaps we want to compare nested models where a reduced model uses a fixed parameter value while the full model estimates the parameter. Fixing spatial covariance parameters while fitting a model is possible using the `spcov_initial` argument to `splm()` and `spautor()`. The `spcov_initial` argument takes an `spcov_initial` object (run `help("spcov_initial", "spmodel")` for more). `spcov_initial` objects can also be used to specify initial values used during optimization, even if they are not assumed to be fixed. By default, `spmodel` uses a grid search to find suitable initial values to use during optimization.

As an example, suppose our goal is to compare a model with an exponential covariance and dependent error variance, independent error variance, and range parameter to a similar model that instead assumes the independent random error variance parameter (nugget) is zero. First, the `spcov_initial` object is specified for the latter model:

```
R> init <- spcov_initial("exponential", ie = 0, known = "ie")
R> init
$initial
ie
 0

$is_known
  ie
TRUE

attr(,"class")
[1] "exponential"
```

The `init` output shows that the `ie` parameter has an initial value of zero that is assumed to be known. Next the model is fit:

```
R> spmod_red <- splm(log_Zn ~ log_dist2road, moss, spcov_initial = init)
```

Notice that because the `spcov_initial` object contains information about the spatial covariance type, the `spcov_type` argument is not required when `spcov_initial` is provided. We can use `glances()` to glance at both models:

```
R> glances(spmod, spmod_red)

# A tibble: 2 x 10
  model         n     p  npar value   AIC  AICc logLik deviance
  <chr>     <int> <dbl> <int> <dbl> <dbl> <dbl>  <dbl>    <dbl>
1 spmod       365     2     3  367.  373.  373.  -184.      363
2 spmod_red   365     2     2  378.  382.  382.  -189.     374.
# ... with 1 more variable: pseudo.r.squared <dbl>
```

The lower AIC and AICc of the full model compared to the reduced model indicates that the independent random error variance is important to the model. A likelihood ratio test comparing the full and reduced models is also possible using `anova()`.

Another application of fixing spatial covariance parameters involves calculating their profile likelihood confidence intervals [38]. Before calculating a profile likelihood confidence

interval for $\Theta_i$, the $i$th element of a general parameter vector $\Theta$, it is necessary to obtain $-2\ell(\hat{\Theta})$, minus twice the log-likelihood evaluated at the estimated parameter vector, $\hat{\Theta}$. Then a $(1 - \alpha)\%$ profile likelihood confidence interval is the set of values for $\Theta_i$ such that $2\ell(\hat{\Theta}) - 2\ell(\hat{\Theta}_{-i}) \leq \chi^2_{1,1-\alpha}$, where $\ell(\hat{\Theta}_{-i})$ is the value of the log-likelihood maximized after fixing $\Theta_i$ and optimizing over the remaining parameters, $\Theta_{-i}$, and $\chi^2_{1,1-\alpha}$ is the $1 - \alpha$ quantile of a chi-squared distribution with one degree of freedom. The result follows from inverting a likelihood ratio test comparing the full model to a reduced model that fixes the value of $\Theta_i$. Because computing profile likelihood confidence intervals requires refitting the model many times for different fixed values of $\Theta_i$, it can be computationally intensive. This approach can be generalized to yield joint profile likelihood confidence intervals cases when $i$ has dimension greater than one.

## Fitting and predicting for multiple models

Fitting multiple models is possible with a single call to `splm()` or `spautor()` when `spcov_type` is a vector with length greater than one or `spcov_initial` is a list (with length greater than one) of `spcov_initial` objects. We fit three separate spatial linear models using the exponential spatial covariance, spherical spatial covariance, and no spatial covariance by running

```
R> spmods <- splm(
+    sulfate ~ 1,
+    sulfate,
+    spcov_type = c("exponential", "spherical", "none")
+ )
```

Then `glances()` is used to glance at each fitted model object:

```
R> glances(spmods)
```

```
# A tibble: 3 x 10
  model           n     p  npar value   AIC  AICc logLik deviance
  <chr>       <int> <dbl> <int> <dbl> <dbl> <dbl>  <dbl>    <dbl>
1 spherical     197     1     3 1137. 1143. 1143.  -569.     196.
2 exponential   197     1     3 1140. 1146. 1146.  -570.     196.
3 none          197     1     1 1448. 1450. 1450.  -724.     196
# ... with 1 more variable: pseudo.r.squared <dbl>
```

And `predict()` is used to predict `newdata` separately fo each fitted model object:

```
R> predict(spmods, newdata = sulfate_preds)
```

Currently, `glances()` and `predict()` are the only `spmodel` generic functions that operate on an object that contains multiple model fits. Generic functions that operate on individual models can still be called when the argument is an individual model object. For example, we can compute the AIC of the model fit using the exponential covariance function by running

```
R> AIC(spmods$exponential)
```

```
[1] 1145.824
```

## Random effects

Non-spatial random effects incorporate additional sources of variability into model fitting. They are accommodated in `spmodel` using similar syntax as for random effects in the nlme [32] and lme4 [39] **R** packages. Random effects are specified via a formula passed to the `random` argument. Next, we show two examples that incorporate random effects into the spatial linear model using the `moss` data.

The first example explores random intercepts for the `sample` variable. The `sample` variable indexes each unique location, which can have replicate observations due to field duplicates (`field_dup`) and lab replicates (`lab_rep`). There are 365 observations in `moss` at 318 unique locations, which means that 47 observations in `moss` are either field duplicates or lab replicates. It is likely that the repeated observations at a location are correlated with one another. We can incorporate this repeated-observation correlation by creating a random intercept for each level of `sample`. We model the random intercepts for `sample` by running

```
R> rand1 <- splm(
+    log_Zn ~ log_dist2road,
+    moss,
+    spcov_type = "exponential",
+    random = ~ sample
+ )
```

Note that `~ sample` is shorthand for `~ (1 | sample)`, which is more explicit notation that indicates random intercepts for each level of `sample`.

The second example adds a random intercept for `year`, which creates extra correlation for observations within a year. It also adds a random slope for `log_dist2road` within `year`, which lets the effect of `log_dist2road` vary between years. We fit this model by running

```
R> rand2 <- splm(
+    log_Zn ~ log_dist2road,
+    moss,
+    spcov_type = "exponential",
+    random = ~ sample + (log_dist2road | year)
+ )
```

Note that `~ sample + (log_dist2road | year)` is shorthand for `~ (1 | sample) + (log_dist2road | year)`. If only random slopes within a year are desired (and no random intercepts), a `- 1` is given to the relevant portion of the formula: `(log_dist2road-1 | year)`. When there is more than one term in `random`, each term must be surrounded by parentheses (recall that the random intercept shorthand automatically includes relevant parentheses).

We can compare the AIC of all three models by running

```
R> AIC(spmod, rand1, rand2)

       df       AIC
spmod   3  373.2089
rand1   4  343.1021
rand2   6  201.8731
```

The `rand2` model has the lowest AIC.

It is possible to fix random effect variances using the `randcov_initial` argument, and `randcov_initial` can also be used to set initial values for optimization.

## Partition factors

A partition factor is a variable that allows observations to be uncorrelated when they are from different levels of the partition factor. Partition factors are specified in `spmodel` by providing a formula with a single variable to the `partition_factor` argument. Suppose that for the `moss` data, we would like observations in different years (`year`) to be uncorrelated. We fit a model that treats year as a partition factor by running

```
R> part <- splm(
+   log_Zn ~ log_dist2road,
+   moss,
+   spcov_type = "exponential",
+   partition_factor = ~ year
+ )
```

## Anisotropy

An isotroptic spatial covariance function (for point-referenced data) behaves similarly in all directions (i.e., is independent of direction) as a function of distance. An anisotropic covariance function does not behave similarly in all directions as a function of distance. Consider the spatial covariance imposed by an eastward-moving wind pattern. A one-unit distance in the x-direction likely means something different than a one-unit distance in the y-direction. Fig 5 shows ellipses for an isotropic and anisotropic covariance function centered at the origin (a distance of zero). The black outline of each ellipse is a level curve of equal correlation. The left ellipse (a circle) represents an isotropic covariance function. The distance at which the correlation between two observations lays on the level curve is the same in all directions. The right ellipse represents an anisotropic covariance function. The distance at which the correlation between two observations lays on the level curve is different in different directions.

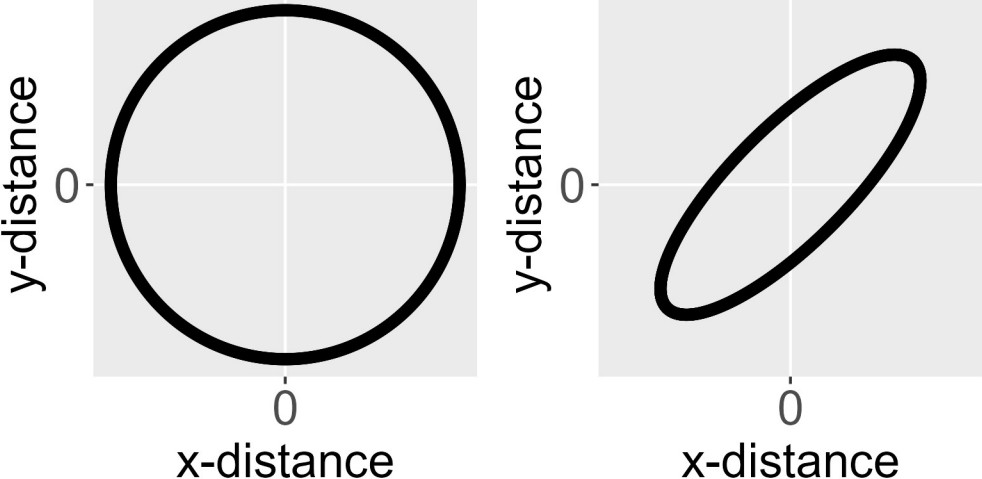

**Fig 5. Ellipses for an isotropic and anisotropic covariancefunction centered at the origin.** In A (left), the isotropic covariance function is visualized. In B (right), the anisotropic covariance function is visualized. The black outline of each ellipse is a level curve of equal correlation.

Accounting for anisotropy involves a rotation and scaling of the x-coordinates and y-coordinates such that the spatial covariance function that uses these transformed distances is isotropic. We use the `anisotropy` argument to `splm()` to fit a model with anisotropy by running

```
R> spmod_anis <- splm(
+   log_Zn ~ log_dist2road,
+   moss,
+   spcov_type = "exponential",
+   anisotropy = TRUE
+ )
R> summary(spmod_anis)

Call:
splm(formula = log_Zn ~ log_dist2road, data = moss, spcov_type = "exponen-
tial",
    anisotropy = TRUE)

Residuals:
    Min      1Q   Median      3Q      Max
-2.5279 -1.2239 -0.7202 -0.1921  1.1659

Coefficients (fixed):
              Estimate Std. Error z value Pr(>|z|)
(Intercept)    9.54798    0.22291   42.83   <2e-16 ***
log_dist2road -0.54601    0.01855  -29.44   <2e-16 ***
---
Signif. codes:  0 '***' 0.001 '**' 0.01 '*' 0.05 '.' 0.1 ' ' 1

Pseudo R-squared: 0.7048

Coefficients (exponential spatial covariance):
      de        ie      range     rotate     scale
3.561e-01 6.812e-02 8.732e+03 2.435e+00 4.753e-01
attr(,"class")
[1] "exponential"
```

The `rotate` parameter is between zero and $\pi$ radians and represents the angle of a clockwise rotation of the ellipse such that the major axis of the ellipse is the new x-axis and the minor axis of the ellipse is the new y-axis. The `scale` parameter is between zero and one and represents the ratio of the distance between the origin and the edge of the ellipse along the minor axis to the distance between the origin and the edge of the ellipse along the major axis. The transformation that turns an anisotropic ellipse into an isotropic one (i.e., a circle) requires rotating the coordinates clockwise by `rotate` and then scaling them the reciprocal of `scale`. The transformed coordinates are then used instead of the original coordinates to compute distances and spatial covariances.

Note that specifying an initial value for `rotate` that is different from zero, specifying an initial value for `scale` that is different from one, or assuming either `rotate` or `scale` are unknown in `spcov_initial` will cause `splm()` to fit a model with anisotropy (and will override `anisotropy = FALSE`). Estimating anisotropy parameters is only possible for

maximum likelihood and restricted maximum likelihood estimation, but fixed anisotropy parameters can be accommodated for semivariogram weighted least squares or semivariogram composite likelihood estimation. Also note that anisotropy is not relevant for areal data because the spatial covariance function depends on a neighborhood structure instead of distances between locations.

## Simulating spatial data

The `sprnorm()` function is used to simulate normal (Gaussian) spatial data. To use `sprnorm()`, the `spcov_params()` function is used to create an `spcov_params` object. The `spcov_params()` function requires the spatial covariance type and parameter values. We create an `spcov_params` object by running

```
R> sim_params <- spcov_params("exponential", de = 5, ie = 1, range = 0.5)
```

We set a reproducible seed and then simulate data at 3000 random locations in the unit square using the spatial covariance parameters in `sim_params` by running

```
R> set.seed(0)
R> n <- 3000
R> x <- runif(n)
R> y <- runif(n)
R> coords <- tibble::tibble(x, y)
R> resp <- sprnorm(
+   sim_params,
+   data = coords,
+   xcoord = x,
+   ycoord = y
+ )
R> sim_data <- tibble::tibble(coords, resp)
```

We can visualize the simulated data (Fig 6A) by running

```
R> ggplot(sim_data, aes(x = x, y = y, color = resp)) +
+   geom_point(size = 1.5) +
+   scale_color_viridis_c(limits = c(-7, 7)) +
+   theme_gray(base_size = 18)
```

There is noticeable spatial patterning in the response variable (`resp`). The default mean in `sprnorm()` is zero for all observations, though a mean vector can be provided using the `mean` argument. The default number of samples generated in `sprnorm()` is one, though this can be changed using the `samples` argument. Because `sim_data` is a `tibble` (`data.frame`) and not an `sf` object, the columns in `sim_data` representing the x-coordinates and y-coordinates must be provided to `sprnorm()`.

Note that the output from `coef(object, type = "spcov")` is a `spcov_params` object. This is useful we want to simulate data given the estimated spatial covariance parameters from a fitted model. Random effects are incorporated into simulation via the `randcov_-params` argument.

## Big data

The computational cost associated with model fitting is exponential in the sample size for all estimation methods. For maximum likelihood and restricted maximum likelihood, the

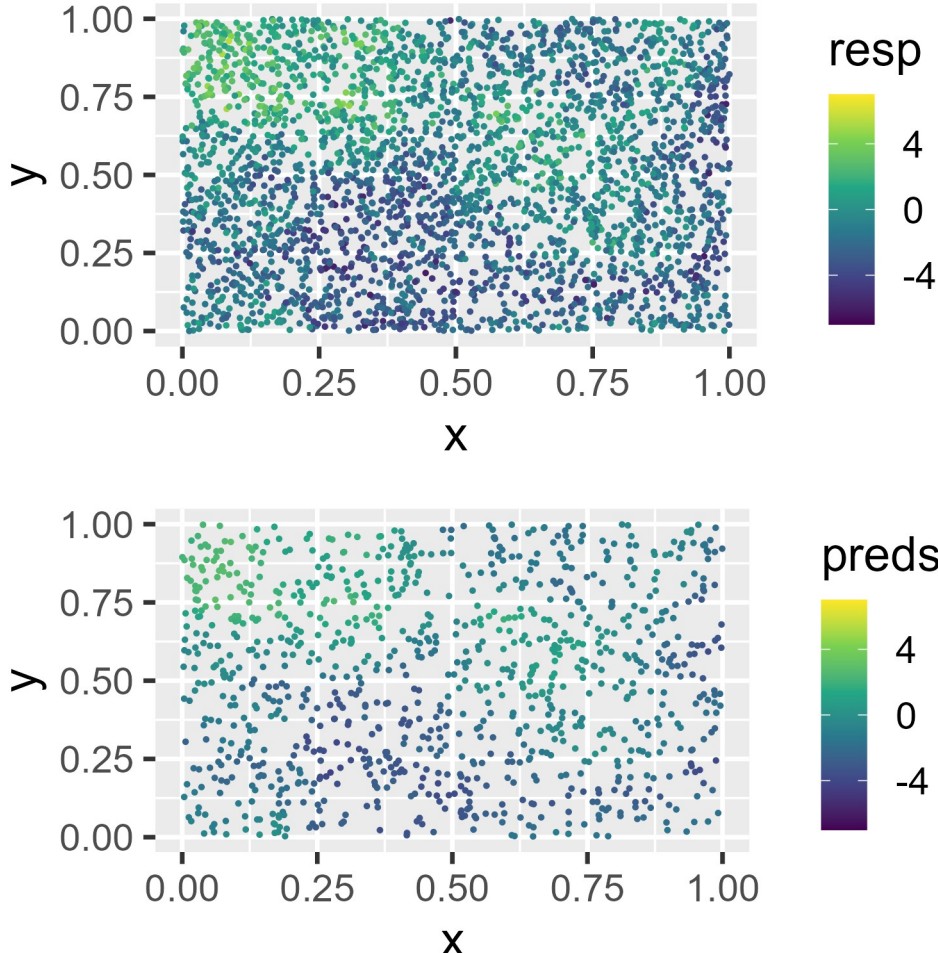

**Fig 6. Observed data and big data predictions at unobserved locations.** In A (top), spatial data are simulated in the unit square. A spatial linear model is fit using the default big data approximation for model-fitting. In B (bottom), predictions are made using the fitted model and the default big data approximation for prediction.

computational cost of estimating $\boldsymbol{\theta}$ is cubic. For semivariogram weighted least squares and semivariogram composite likelihood, the computational cost of estimating $\boldsymbol{\theta}$ is quadratic. The computational cost associated with estimating $\boldsymbol{\beta}$ and prediction is cubic in the model-fitting sample size, regardless of estimation method. Typically, samples sizes approaching 10,000 make the computational cost of model fitting and prediction infeasible, which necessitates the use of big data methods. `spmodel` offers big data methods for model fitting of point-referenced data via the `local` argument to `splm()`. The method is capable of quickly fitting models with hundreds of thousands to millions of observations. Because of the neighborhood structure of areal data, the big data methods used for point-referenced data do not apply to areal data. Thus, there is no big data method for areal data or `local` argument to `spautor ()`, so model fitting sample sizes cannot be too large. `spmodel` offers big data methods for prediction of point-referenced data or areal data via the `local` argument to `predict()`, capable of quickly predicting hundreds of thousands to millions of observations rather quickly.

To show how to use `spmodel` for big data estimation and prediction, we use the `sim_ data` data from the previous subsection. Because `sim_data` is a `tibble` (`data.frame`)

and not an `sf` object, the columns in `data` representing the x-coordinates and y-coordinates must be explicitly provided to `splm()`.

**Model-fitting.** `spmodel` uses a "local indexing" approximation for big data model fitting of point-referenced data. Observations are first assigned an index. Then for the purposes of model fitting, observations with different indexes are assumed uncorrelated. Assuming observations with different indexes are uncorrelated induces sparsity in the covariance matrix, which greatly reduces the computational time of operations that involve the covariance matrix.

The `local` argument to `splm()` controls the big data options. `local` is a list with several arguments. The arguments to the `local` list control the method used to assign the indexes, the number of observations with the same index, the number of unique indexes, adjustments to the covariance matrix of $\hat{\boldsymbol{\beta}}$, whether or not to use parallel processing, and if parallel processing is used, the number of cores.

Big data are most simply accommodated by setting `local` to `TRUE`. This is shorthand for `local = list(method = "random", size = 50, var_adjust = "theoretical", parallel = FALSE)`, which randomly assigns observations to index groups, ensures each index group has approximately 50 observations, uses the theoretically-correct covariance adjustment, and does not use parallel processing.

```
R> local1 <- splm(
+   resp ~ 1,
+   sim_data,
+   spcov_type = "exponential",
+   xcoord = x,
+   ycoord = y,
+   local = TRUE
+ )
R> summary(local1)

Call:
splm(formula = resp ~ 1, data = sim_data, spcov_type = "exponential",
    xcoord = x, ycoord = y, local = TRUE)

Residuals:
    Min      1Q  Median      3Q     Max
-5.0356 -1.3514 -0.1468  1.2842  6.5381

Coefficients (fixed):
            Estimate Std. Error z value Pr(>|z|)
(Intercept)   -1.021      0.699   -1.46    0.144

Coefficients (exponential spatial covariance):
    de     ie  range
2.8724 0.9735 0.2644
```

Instead of using `local = TRUE`, we can explicitly set `local`. For example, we can fit a model using k-means clustering [40] on the x-coordinates and y-coordinates to create 60 groups (clusters), use the pooled variance adjustment, and use parallel processing with two cores by running

```
R> local2_list <- list(
+    method = "kmeans",
+    groups = 60,
+    var_adjust = "pooled",
+    parallel = TRUE,
+    ncores = 2
+ )
R> local2 <- splm(
+    resp ~ 1,
+    sim_data,
+    spcov_type = "exponential",
+    xcoord = x,
+    ycoord = y,
+    local = local2_list
+ )
R> summary(local2)

Call:
splm(formula = resp ~ 1, data = sim_data, spcov_type = "exponential",
     xcoord = x, ycoord = y, local = local2_list)

Residuals:
     Min        1Q    Median        3Q       Max
-4.98801  -1.30386  -0.09927   1.33176   6.58567

Coefficients (fixed):
             Estimate Std. Error z value Pr(>|z|)
(Intercept)   -1.0683     0.1759  -6.073 1.25e-09 ***
---
Signif. codes:  0 '***' 0.001 '**' 0.01 '*' 0.05 '.' 0.1 ' ' 1

Coefficients (exponential spatial covariance):
    de      ie   range
2.5434  0.9907  0.2312
```

Likelihood-based statistics like `AIC()`, `AICc()`, `logLik()`, and `deviance()` should not be used to compare a model fit with a big data approximation to a model fit without a big data approximation, as the two approaches maximize different likelihoods.

**Prediction.** For point-referenced data, `spmodel` uses a "local neighborhood" approximation for big data prediction. Each prediction is computed using a subset of the observed data instead of all of the observed data. Before further discussing big data prediction, we simulate 1000 locations in the unit square requiring prediction:

```
R> n_pred <- 1000
R> x <- runif(n_pred)
R> y <- runif(n_pred)
R> sim_preds <- tibble::tibble(x = x, y = y)
```

The `local` argument to `predict()` controls the big data options. `local` is a list with several arguments. The arguments to the `local` list control the method used to subset the observed data, the number of observations in each subset, whether or not to use parallel processing, and if parallel processing is used, the number of cores.

The simplest way to accommodate big data prediction is to set `local` to `TRUE`. This is shorthand for `local = list(method = "covariance", size = 50, parallel = FALSE)`, which implies that for each location requiring prediction, only the 50 observations in the data most correlated with it are used in the computation, and parallel processing is not used. Using the `local1` fitted model, we store these predictions as a variable called `preds` in the `sim_preds` data by running

```
R> sim_preds$preds <- predict(local1, newdata = sim_preds, local = TRUE)
```

The predictions are visualized (Fig 6B) by running

```
R> ggplot(sim_preds, aes(x = x, y = y, color = preds)) +
+   geom_point(size = 1.5) +
+   scale_color_viridis_c(limits = c(-7, 7)) +
+   theme_gray(base_size = 18)
```

They display a similar pattern as the observed data.

Instead of using `local = TRUE`, we can explicitly set `local`:

```
R> pred_list <- list(
+   method = "distance",
+   size = 30,
+   parallel = TRUE,
+   ncores = 2
+ )
R> predict(local1, newdata = sim_preds, local = pred_list)
```

This code implies that uniquely for each location requiring prediction, only the 30 observations in the data closest to it (in terms of Euclidean distance) are used in the computation and parallel processing is used with two cores.

For areal data, no local neighborhood approximation exists because of the data's underlying neighborhood structure. Thus, all of the data must be used to compute predictions and by consequence, `method` and `size` are not components of the `local` list. The only components of the `local` list for areal data are `parallel` and `ncores`.

## Discussion

`spmodel` is a novel, relevant contribution used to fit, summarize, and predict for a variety of spatial statistical models. Spatial linear models for point-referenced data (i.e., geostatistical models) are fit using the `splm()` function. Spatial linear models for areal data (i.e., autoregressive models) are fit using the `spautor()` function. Both functions use a common framework and syntax structure. Several model-fit statistics and diagnostics are available. The broom functions `tidy()` and `glance()` are used to tidy and glance at a fitted model. The broom function `augment()` is used to augment `data` with model diagnostics and augment `newdata` with predictions. Several advanced features are available to accommodate fixed covariance parameter values, random effects, partition factors, anisotropy, simulating data, and big data approximations for model fitting and prediction.

We appreciate feedback from users regarding `spmodel`, and we have several plans to add new features to `spmodel` in the future. To learn more about `spmodel` or provide feedback, please visit our website at https://usepa.github.io/spmodel/.

## Acknowledgments

We would like to thank the editor and anonymous reviewers for their feedback which greatly improved the manuscript.

The views expressed in this manuscript are those of the authors and do not necessarily represent the views or policies of the U.S. Environmental Protection Agency or the National Oceanic and Atmospheric Administration. Any mention of trade names, products, or services does not imply an endorsement by the U.S. government, the U.S. Environmental Protection Agency, or the National Oceanic and Atmospheric Administration. The U.S. Environmental Protection Agency and the National Oceanic and Atmospheric Administration do not endorse any commercial products, services or enterprises.

## Author Contributions

**Conceptualization:** Michael Dumelle.

**Data curation:** Michael Dumelle, Jay M. Ver Hoef.

**Formal analysis:** Michael Dumelle, Matt Higham, Jay M. Ver Hoef.

**Investigation:** Michael Dumelle, Matt Higham, Jay M. Ver Hoef.

**Methodology:** Michael Dumelle, Matt Higham, Jay M. Ver Hoef.

**Project administration:** Michael Dumelle, Matt Higham, Jay M. Ver Hoef.

**Resources:** Michael Dumelle, Matt Higham, Jay M. Ver Hoef.

**Software:** Michael Dumelle, Matt Higham, Jay M. Ver Hoef.

**Supervision:** Michael Dumelle, Matt Higham, Jay M. Ver Hoef.

**Validation:** Michael Dumelle, Matt Higham, Jay M. Ver Hoef.

**Visualization:** Michael Dumelle, Matt Higham, Jay M. Ver Hoef.

**Writing – original draft:** Michael Dumelle, Matt Higham, Jay M. Ver Hoef.

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
