## [Decision Letter · Decision Letter 0]

11 Jan 2023

PONE-D-22-28137spmodel: spatial statistical modeling and prediction in RPLOS ONE

Dear Dr. Dumelle,

Thank you for submitting your manuscript to PLOS ONE. After careful consideration, we feel that it has merit but does not fully meet PLOS ONE’s publication criteria as it currently stands. Therefore, we invite you to submit a revised version of the manuscript that addresses the points raised during the review process.

Kind regards,

A. K. M. Anisur Rahman, Ph.D.

Academic Editor

PLOS ONE

Journal Requirements:

Reviewers' comments:

Reviewer's Responses to Questions

**Comments to the Author**

1. Is the manuscript technically sound, and do the data support the conclusions?

Reviewer #1: Yes

Reviewer #2: Yes

2. Has the statistical analysis been performed appropriately and rigorously? 

Reviewer #1: Yes

Reviewer #2: Yes

3. Have the authors made all data underlying the findings in their manuscript fully available?

Reviewer #1: Yes

Reviewer #2: Yes

4. Is the manuscript presented in an intelligible fashion and written in standard English?

Reviewer #1: Yes

Reviewer #2: Yes

5. Review Comments to the Author

Reviewer #1: sp: spatial statistical modelling and prediction in R

PONE-D-22-28137

General comments

It was a pleasure to review this manuscript. The authors clearly and concisely described the functionality of their R package, spmodel, and provided well-considered examples. I was impressed with the breadth of functionality that spmodel provides including spatial autoregressive and geostatistical models, anisotrophy, prediction, random effects, partitioning, simulation, and functions to handle modelling and prediction of big data. There are a few packages that provide a subset of this functionality, but I don’t know of any that provide all of it. The syntax is similar to lm(), which will make it familiar to almost anyone who has used R and the inclusion of the broom functions make it intuitive to everyone familiar with tidy. The paper provides a nice balance of showing the package functionality and explaining how to interpret the outputs (effective range, AIC, etc.), making it accessible to readers with different levels of statistical experience. I was very excited to see a few examples of interactive maps in the paper and have no doubt that readers will want to recreate these visualisations. I was able to run all of the R code without error and all of the helpfiles I looked at were informative and complete. I didn’t review the technical details document, but I appreciate that the authors provided that additional resource. I’ve made some minor suggestions below about how to improve the manuscript, but my recommendation is to accept.

Minor suggestions:

Data are always plural. Change ‘data is’ to ‘data are’ throughout, with the exception of the object named ‘data’.

Introduction: Suggest mentioning INLA in review of R packages used for modelling point-referenced and areal data. I believe INLA and rstan can be used to model both, but in both cases the unique object structure and syntax leads to a fairly significant learning curve for new users.

Lines 238-239: I suggest reminding the reader that both models are fit using the default estimation method, REML, and so AIC and AICc are both valid for model comparison. Alternatively, you could add the estmethod = "reml" to the function call to make it more obvious.

Line 271: This sentence about leverage feels incomplete compared to the explanation given to other functions and outputs. I suggest adding something to the effect of ‘large leverage values indicate that an observation may be an outlier and warrant further investigation’, without going into too much detail. Later I see that leverage is mentioned again in lines 302-303 and Cooks D vs leverage is plotted. Another alternative is to add a sentence here telling readers how to interpret the plot.

Lines 274-276: I suggest including an example call to help here, which users will need to do if they’d like to change the type in the fitted function. I had trouble figuring out that the function was named fitted.spmod. I may have missed it in the help file and manuscript, but I don’t see where it says the output of splm is an object of class “spmod” – that would have helped here.

Line 296, 317, 388, 399, 558, 647, 697: Unindent so that it is part of the previous paragraph.

Lines 412-443. – output from spautor: It would be helpful to add a sentence or three about the output. For example, where is the autocorrelation parameter estimate? I noticed that the pseudo.r.squared is 0. Am I correct in my interpretation that this model has a really poor model fit? Or does it return 0 because there are no covariates in the model?

Lines 471-478: And the same factor levels if variables are factors?

Predict function: It looks like the results are returned in different formats (vector, list), depending on the se.fit and interval arguments. It would be worthwhile to mention this here so that people know to refer to the help file, which explains this clearly. I also suggest including the example with augment (lines 495-500) in the help file, as well as the manuscript. It’s really helpful.

Line 667: then is used twice in this sentence – awkward.

Random Effects section: I really appreciate all of the examples the authors included and the additional descriptions they’ve provided about how to set up random effects. This is going to be very helpful for users who are either unfamiliar with R or come from a non-statistical background.

Line 724: Incomplete sentence.

Line 810: typo – ‘and’ is repeated twice.

Lines 869-884: Are the indices mentioned in lines 869-871 used to define the groups mentioned in lines 880-883? I assume that the local parameter list defines the groups and the indices are the labels?

Reviewer #2: This paper describes the main functionalities of the R package “spmodel”, designed to analyze point-referenced and areal data using a common framework and syntax structure. The package allows to fit several geostatistical models for point-referenced data through the splm() function, and both SAR and CAR models for spatial areal data through the spautor() function. Different estimation methods are also available (restricted maximum likelihood, maximum likelihood, semivariogram weighted least squares and semivariogram composite likelihood) to obtain point estimates of the fixed effects and covariance parameters of the spatial linear model. Finally, the paper describes additional features of the package such as model-fit statistics, diagnostic metrics and spatial interpolation (or Kriging) among others.

The manuscript is well written and clearly describes the main functionalities of the package, including as supplementary material the data and R code to reproduce the results shown in the paper.

My major concerns are described below:

1) Redaction style:

Clearly, the manuscript is written following the style of a paper submitted to “Journal of Statistical Software” or “The R Journal”. I am not member of the Editorial Board of PLOS ONE, so I do not see myself qualified to judge whether the current style of the manuscript is appropriate to be published in this journal.

2) Introduction:

Very few references are included in the first part of the introduction section and most of them are packages/papers written by the authors of the present manuscript. Please, include additional references to spatial random sampling and statistical analysis of spatial data (as for example, [1]).

When reviewing the existing R packages to analyze and estimate areal data, I suggest the authors to include also the “diseasemapping” [2] and “bigDM” [3] packages. Additional packages for disease mapping and areal data analysis can be found in the CRAN Task View “Analysis of Spatial Data” (https://cran.r-project.org/web/views/Spatial.html)

3) Additional comment:

Perhaps the authors want to update the manuscript by including some of the new features from the current version (0.2.0) of the “spmodel” package.

References:

[1] Banerjee, S., Carlin, B. P., & Gelfand, A. E. (2003). Hierarchical Modeling and Analysis for Spatial Data. Chapman and Hall/CRC.

[2] Brown, P. E. (2015). Model-based geostatistics the easy way. Journal of Statistical Software, 63, 1-24.

[3] Adin, A., Orozco-Acosta, E., and Ugarte, M.D. (2022). bigDM: scalable Bayesian disease mapping models for high-dimensional data. https://CRAN.R-project.org/package=bigDM

6. PLOS authors have the option to publish the peer review history of their article (what does this mean?). If published, this will include your full peer review and any attached files.

Reviewer #1: No

Reviewer #2: No

While revising your submission, please upload your figure files to the Preflight Analysis and Conversion Engine (PACE) digital diagnostic tool, https://pacev2.apexcovantage.com/. PACE helps ensure that figures meet PLOS requirements. To use PACE, you must first register as a user. Registration is free. Then, login and navigate to the UPLOAD tab, where you will find detailed instructions on how to use the tool. If you encounter any issues or have any questions when using PACE, please email PLOS at figures@plos.org. Please note that Supporting Information files do not need this step.<quillbot-extension-portal></quillbot-extension-portal>

---

## [Author Response · Author response to Decision Letter 0]

26 Jan 2023

Please see the attached review_response.pdf for all of the relevant review comments. Thanks.

---

## [Decision Letter · Decision Letter 1]

13 Feb 2023

PONE-D-22-28137R1spmodel: spatial statistical modeling and prediction in RPLOS ONE

Dear Dr. Dumelle,

Thank you for submitting your manuscript to PLOS ONE. After careful consideration, we feel that it has merit but does not fully meet PLOS ONE’s publication criteria as it currently stands. Therefore, we invite you to submit a revised version of the manuscript that addresses the points raised during the review process.

Kind regards,

A. K. M. Anisur Rahman, Ph.D.

Academic Editor

PLOS ONE

Journal Requirements:

Reviewers' comments:

Reviewer's Responses to Questions

**Comments to the Author**

1. If the authors have adequately addressed your comments raised in a previous round of review and you feel that this manuscript is now acceptable for publication, you may indicate that here to bypass the “Comments to the Author” section, enter your conflict of interest statement in the “Confidential to Editor” section, and submit your "Accept" recommendation.

Reviewer #1: All comments have been addressed

Reviewer #2: (No Response)

2. Is the manuscript technically sound, and do the data support the conclusions?

Reviewer #1: (No Response)

Reviewer #2: Yes

3. Has the statistical analysis been performed appropriately and rigorously? 

Reviewer #1: (No Response)

Reviewer #2: Yes

4. Have the authors made all data underlying the findings in their manuscript fully available?

Reviewer #1: (No Response)

Reviewer #2: Yes

5. Is the manuscript presented in an intelligible fashion and written in standard English?

Reviewer #1: (No Response)

Reviewer #2: Yes

6. Review Comments to the Author

Reviewer #1: (No Response)

Reviewer #2: The authors have made a comprehensive revision of the present manuscript and all my comments have been addressed.

I have no substantive comments, just a couple of small details that I describe below:

1) Page 2, first paragraph: Please, correct the name of the R-INLA package (instead of R-inla) and use the same letter font for the bigDM package

2) References section: Check the capital letters in the titles/journal names of some references such as [11], [16] or [40]

7. PLOS authors have the option to publish the peer review history of their article (what does this mean?). If published, this will include your full peer review and any attached files.

Reviewer #1: No

Reviewer #2: No

While revising your submission, please upload your figure files to the Preflight Analysis and Conversion Engine (PACE) digital diagnostic tool, https://pacev2.apexcovantage.com/. PACE helps ensure that figures meet PLOS requirements. To use PACE, you must first register as a user. Registration is free. Then, login and navigate to the UPLOAD tab, where you will find detailed instructions on how to use the tool. If you encounter any issues or have any questions when using PACE, please email PLOS at figures@plos.org. Please note that Supporting Information files do not need this step.<quillbot-extension-portal></quillbot-extension-portal>

---

## [Author Response · Author response to Decision Letter 1]

14 Feb 2023

I have attached relevant responses to the reviewer comments in the review_responses.pdf document attached as part of this submission.

---

## [Editor Report · Decision Letter 2]

17 Feb 2023

spmodel: spatial statistical modeling and prediction in R

PONE-D-22-28137R2

Dear Dr. Dumelle,

We’re pleased to inform you that your manuscript has been judged scientifically suitable for publication and will be formally accepted for publication once it meets all outstanding technical requirements.

Kind regards,

A. K. M. Anisur Rahman, Ph.D.

Academic Editor

PLOS ONE

Additional Editor Comments (optional):

Reviewers' comments:

<quillbot-extension-portal></quillbot-extension-portal>

---

## [Editor Report · Acceptance letter]

27 Feb 2023

PONE-D-22-28137R2 

spmodel: spatial statistical modeling and prediction in R 

Dear Dr. Dumelle:

I'm pleased to inform you that your manuscript has been deemed suitable for publication in PLOS ONE. Congratulations! Your manuscript is now with our production department. 

Kind regards, 

on behalf of

Dr. A. K. M. Anisur Rahman 

Academic Editor

PLOS ONE